# CAPSTONE: Curriculum Sampling for Dense Retrieval with Document Expansion

**Xingwei He**[1,*],  **Yeyun Gong**[2,†],  **A-Long Jin**[1],  **Hang Zhang**[2],
**Anlei Dong**[3],  **Jian Jiao**[3],  **Siu Ming Yiu**[1,†],  **Nan Duan**[2]

[1]The University of Hong Kong, [2]Microsoft Research Asia, [3]Microsoft

hexingwei15@gmail.com, ajin@eee.hku.hk, smyiu@cs.hku.hk,
{yegong, v-zhhang, anlei.dong, jian.jiao, nanduan}@microsoft.com

## Abstract

The dual-encoder has become the *de facto* architecture for dense retrieval. Typically, it computes the latent representations of the query and document independently, thus failing to fully capture the interactions between the query and document. To alleviate this, recent research has focused on obtaining query-informed document representations. During training, it expands the document with a real query, but during inference, it replaces the real query with a generated one. This inconsistency between training and inference causes the dense retrieval model to prioritize query information while disregarding the document when computing the document representation. Consequently, it performs even worse than the vanilla dense retrieval model because its performance heavily relies on the relevance between the generated queries and the real query. In this paper, we propose a curriculum sampling strategy that utilizes pseudo queries during training and progressively enhances the relevance between the generated query and the real query. By doing so, the retrieval model learns to extend its attention from the document alone to both the document and query, resulting in high-quality query-informed document representations. Experimental results on both in-domain and out-of-domain datasets demonstrate that our approach outperforms previous dense retrieval models.

## 1 Introduction

Text retrieval aims to find the relevant documents for a given query from a large collection of documents, playing an indispensable role in open-domain question answering (Chen et al., 2017), fact verification (Thorne et al., 2018) and retrieval-augmented generation (Lewis et al., 2020; He et al., 2022). At the early stage, sparse retrieval methods such as TF-IDF or BM25 dominated passage retrieval by relying mainly on lexical term matching

to compute relevance between the query and document. Recently, there has been a surge of research interest in neural network-based dense retrieval (Karpukhin et al., 2020; Xiong et al., 2021). Different from sparse retrieval, dense retrieval resorts to neural encoders to compute the dense representations of the query and document. This enables dense retrieval to infer the relevance between them at the semantic level rather than the surface level, thus circumventing the term mismatch problem suffered by the sparse retrieval models.

In recent years, the dual-encoder architecture has been a standard workhorse for dense retrieval. One major disadvantage of this architecture is that it can only partially extract the interactions between the query and document, since it encodes them separately. By comparison, the cross-encoder architecture can effectively capture the deep correlation between them by taking the concatenation of the query and document as input. By directly concatenating the query and document, the cross-encoder gains an advantage in capturing interactions, but also loses the advantage of pre-computing document representations during inference. Therefore, cross-encoder cannot wholly replace dual-encoder.

To enhance the retrieval models' ability to capture interactions between queries and documents while maintaining retrieval efficiency, previous work mainly focuses on generating query-informed document representations. One approach, known as late interaction (Khattab and Zaharia, 2020), involves encoding the query and document independently in the early layers, while the later layers model their interactions. Late interaction combines dual-encoder and cross-encoder, making a trade-off between retrieval efficiency and performance. On the other hand, Li et al. (2022) proposed a promising retrieval architecture, dual-cross-encoder. As shown in Figure 1 (c), this architecture computes the query-related document representation by expanding the document with a real or pseudo query.

---

*Work done during internship at Microsoft Research Asia.
†Corresponding authors.

Compared with late interaction, dual-cross-encoder (i.e., dense retrieval with document expansion) gets the query-related document representation without sacrificing the retrieval efficiency at inference. However, there exists a discrepancy between training and inference in the current dual-cross-encoder retriever. Specifically, during training, the document is expanded using a real query, whereas during inference, the document is enriched with a generated query. This discrepancy causes the learned retriever overly focus on the query, yet neglect the document, when computing the document representation. During inference, if the generated query $q\prime$ significantly differs from the user-input query $q$, the query-related document representation will be misled by $q\prime$, thus degrading the performance. That is why the dual-cross-encoder even underperforms the vanilla dual-encoder. To address this issue, Li et al. (2022) proposed a solution by computing multiview document representations using different generated queries for each document. While multiview document representations improve retrieval performance, they come at the cost of significantly increased retrieval latency, which scales linearly with the number of views.

In this paper, we propose **CAPSTONE**, a **c**urriculum s**a**m**p**ling for dense re**t**rieval with d**o**cume**n**t **e**xpansion, to bridge the gap between training and inference for dual-cross-encoder. Our motivation is to expect the dual-cross-encoder retrieval model can utilize both the document $d$ and pseudo query $q\prime$ to compute the query-informed document representation. To achieve this, we train the dual-cross-encoder retriever by gradually increasing the relevance of the pseudo query $q\prime$ to the gold query $q$. Specifically, at the early training stage, a pseudo query $q\prime$ irrelevant to $q$ is selected, causing the retriever to solely rely on the document. As we progress to the late training stage, a highly related pseudo query $q\prime$ is chosen, allowing the retriever to learn to augment the document representation with the pseudo query. By doing so, we alleviate the discrepancy between training and inference. During inference, if the user-input query $q$ is similar to the pseudo query $q\prime$, then $q\prime$ will contribute more to making the target document $d$ be retrieved. Otherwise, the retrieval model will mainly rely on the relevance between $q$ and $d$.

To summarize, the main contributions of this paper are as follows: (1) We propose a curriculum learning approach to bridge the gap between train-

ing and inference for dense retrieval with document expansion, further improving the query-informed document representation[1]. (2) We propose to compute the typical document representation rather than using multiview document representations at inference, which balances the retrieval efficiency and performance. (3) To verify the effectiveness, we apply our proposed approach to two different dense retrieval models, DPR (Karpukhin et al., 2020) and coCondenser, and conduct extensive experiments on three in-domain retrieval datasets, and the zero-shot BEIR benchmark. Experimental results show our proposed approach brings substantial gains over competitive baselines. (4) To the best of our knowledge, this is the first time that document expansion is successfully applied to dense retrieval without incurring extra retrieval latency.

## 2 Preliminary

In this section, we introduce the definition of text retrieval and three architectures for dense retrieval.

**Task Description.** Text retrieval is meant to find the most relevant $M$ documents $D^+ = \{d_1^+, d_2^+, \ldots, d_M^+\}$ for the given query $q$ from a large corpus $D = \{d_1, d_2, \ldots, d_N\}$ with $N$ documents ($M \ll N$).

**Dual-Encoder Architecture.** Dual-encoder (DE) is the typical dense retrieval architecture. As shown in Figure 1 (a), it consists of a query encoder $E_q$ and a document encoder $E_d$, which encode the query $q$ and document $d$ into dense vectors $E_q(q)$ and $E_d(d)$, respectively. Previous works usually initialize the encoders with BERT and use the [CLS] vector at the last layer as the dense representation. The similarity score between $q$ and $d$ is measured with the inner product of their vectors:

$$sim(q, d) = E_q(q)^T E_d(d). \quad (1)$$

**Cross-Encoder Architecture.** Since DE models $q$ and $d$ separately, it is not good at capturing the relevance between them. To capture the interactions between them, cross-encoder (CE) directly takes the concatenation of $q$ and $d$ as input, as shown in Figure 1 (b). It first computes the dense representation $E(q + d)$ for them and then uses the fully

---

[1]Our code is available at: https://github.com/microsoft/SimXNS/CAPSTONE.

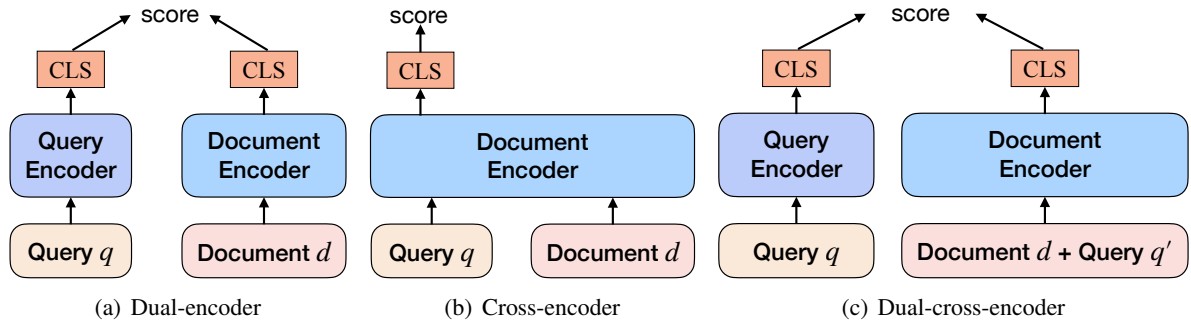

(a) Dual-encoder      (b) Cross-encoder      (c) Dual-cross-encoder

Figure 1: Illustration of the dual-encoder, cross-encoder and dual-cross-encoder architectures. $q$ and $q\prime$ denote the gold query and generated query for the document $d$. '+' is the concatenation operation.

connected layers (FCL) to compute the similarity score:

$$sim(q,d) = FCL(E(d + q)), \qquad (2)$$

where '+' is the concatenation operation.

**Dual-Cross-Encoder Architecture.** Although CE can extract more fine-grained relevance between $q$ and $d$, it is not suitable for retrieval. To combine the advantages of DE and CE, Li et al. (2022) proposed dual-cross-encoder (DCE). As shown in Figure 1 (c), DCE also consists of a query encoder and a document encoder. The only difference between DE and DCE is that the document encoder of DCE takes the document and the generated query $q\prime$ as input during inference. Therefore, DCE can be regarded as DE with document expansion. Similar to DE, DCE computes the similarity score between $q$ and $d$ with the inner product:

$$sim(q,d) = E_q(q)^T E_d(d + q\prime). \qquad (3)$$

## 3 Methodology

In Section 3.1, we first identify the discrepancy between training and inference of DCE. Then, we introduce how to bridge the gap with curriculum learning in Section 3.2. Finally, we will show our proposed inference method in Section 3.3.

### 3.1 Discrepancy in Training and Inference

The training objective of DE is to learn dense representations of queries and documents to maximize the similarity score between the query and positive document. The training loss is defined as follows:

$$L(q, d^+, D^-)$$
$$= -log \frac{e^{sim(q,d^+)}}{e^{sim(q,d^+)} + \sum_{d^- \in D^-} e^{sim(q,d^-)}},$$

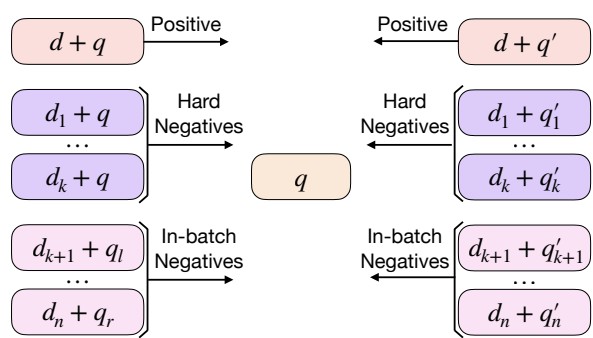

Figure 2: Comparison of different document expansion methods when training DCE, where the left is the vanilla method and the right is our proposed method. $d$ is the positive document for $q$. $d_1, \ldots, d_k$ are the hard negatives. $d_{k+1}, \ldots, d_n$ are the in-batch negatives. $q\prime$ and $q'_i$ are the generated query for $d$ and $d_i$.

where $D^-$ is the set of negative documents, containing hard negatives and in-batch negatives.

During training, DCE expands the positive document $d$ and hard negatives with the gold query $q$ (e.g., replacing $d$ with $d + q$). For ease of understanding, we show how to construct positive, hard negatives and in-batch negatives in the left of Figure 2. As shown in this figure, DCE can filter out all in-batch negatives by solely relying on the query information from the expanded document, since it only needs to observe whether the input query *appears* in the expanded document. The utilization of document information is necessary only for distinguishing the positive from the hard negatives. Consequently, the learned representation of the expanded document has a strong bias toward the query, while almost neglecting the document.

At inference, DCE enriches the document with the generated query $q'$ rather than the user-input query $q$. When $q$ and $q'$ are different types of queries for the target document $d$, $q$ or $E_q(q)$ will be far from $q'$ or $E_d(q')$. As the retrieval model

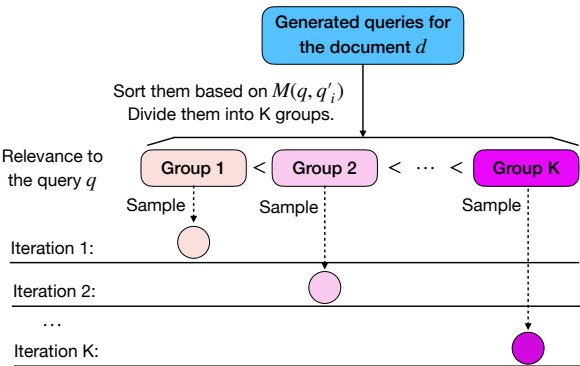

Figure 3: The query selection process based on curriculum learning in each training iteration. We first compute the relevance score $M(q, q'_i)$ between the generated query $q'_i$ and the gold query $q$, and then sort the generated queries in ascending order according to their relevance scores to $q$. Next, we divide them into K groups. At the $i$-th training iteration, we randomly sample one generated query from the $i$-th group, and then use the sampled query to update the dual-cross encoder.

overly relies on the query part when computing the document representation, $E_d(d + q')$ will be close to $E_d(q')$. Based on these, we can easily infer that $E_d(d + q')$ is far away from $E_q(q)$, making it more difficult to retrieve the target document $d$. As a result, the dense retriever with document expansion even underperforms its counterpart without using document expansion. We attribute the performance degradation to the discrepancy between training and inference.

## 3.2 Bridging the Gap with Curriculum Sampling

To bridge the gap between training and inference, we propose to expand the document with its generated query during training. We show the training process of our proposed method in the right of Figure 2. Although this idea is simple and intuitive, it brings a significant advantage: the retriever cannot take a shortcut during training, where the retriever cannot exclude in-batch negatives by simply checking whether the user-input query *appears* in the expanded document.

However, this is not the end of the story. If the generated query $q'$ used to expand $d$ is irrelevant to the gold query $q$, the document retriever of DCE will ignore $q'$, which means DCE will degenerate into DE. On the other hand, if $q'$ and $q$ are similar, we will encounter the abovementioned problem (DCE will overly depend on $q'$). We expect the dense retriever with document expansion can use

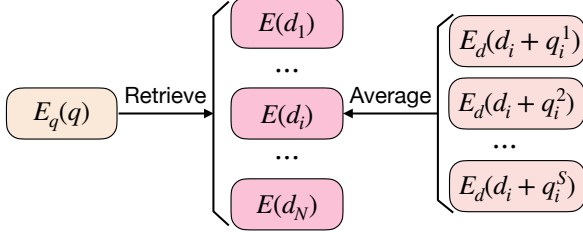

Figure 4: The overview of the retrieval stage. $d_i$ is the $i - th$ document, and $N$ refers to the size of the external corpus. $q^j_i$ means the $j$-th generated query for $d_i$.

both the document and the generated query during training, so that the learned document representation contains the information of both parts.

To fulfill this goal, we further propose the curriculum sampling. To be concrete, at the early training stage, the selected $q\prime$ has low correlations with $q$, forcing the retriever to use the document. As the training goes on, we select $q\prime$ with gradually increased relevance to $q$, encouraging the retriever to use the query. Driven by this motivation, we first generate some queries for each document $d$. Then, we compute the relevance score between the generated query $q'_i$ and the gold query $q$ with an automatic evaluation metric $M$ (i.e., ROUGE-L[2] ). After that, we sort the generated queries in ascending order according to their relevance scores to $q$ and divide them into $K$ groups. We summarize the curriculum sampling in Figure 3.

## 3.3 Inference

Before retrieving, we first generate $S$ queries for each document with the generator. Next, we concatenate a document with one generated query and compute its latent representation with the document encoder. Then, we will get $S$ different representations for each document.

**Corpus Expansion.** Following Li et al. (2022), we keep $S$ representations for each document, so the original corpus will be expanded to $S$ times. When a query comes, we need to retrieve the relevant documents from the expanded corpus, which increases the retrieval latency by $S$ times.

**Computing the Typical Representation of Different Views.** To avoid increasing the retrieval latency, we can retain a single representation for each document. The easiest way is to set $S$ to 1. However, we found that the retrieval performance

---

[2]The evaluation code for ROUGE-L is available at: https://huggingface.co/spaces/evaluate-metric/rouge.

| Datasets | Train | Dev | Test | #Passage |
|----------|-------|-----|------|----------|
| MS-MARCO | 502,939 | 6,980 | - | 8,841,823 |
| TREC-19 | - | - | 43 | 8,841,823 |
| TREC-20 | - | - | 54 | 8,841,823 |

Table 1: Basic statistics of retrieval datasets. #Passage is the number of unique passages in the retrieval corpus.

is positively correlated to the number of views, $S$. This is because a larger value of $S$ increases the probability of the user-input query being relevant to one of the generated queries. Therefore, setting $S$ to 1 will degrade the retrieval performance compared with expanding the corpus.

Given this, we propose to compute the typical representation for a document by averaging all $S$ different views. This approach allows us to maintain only one representation for each document. As shown in Figure 4, our proposed method does not bring additional retrieval latency[3], yet it performs much better than simply setting $S$ to 1 (please refer to Section 4.3 for more details).

## 4 Experiments

### 4.1 Experimental Setups

**Datasets.** We conduct experiments on three passage retrieval datasets: MS-MARCO passage ranking (Bajaj et al., 2016), TREC Deep Learning (DL) Track 2019 (Craswell et al., 2020) and 2020 (Craswell et al., 2021). MS-MARCO collects real user queries from Bing search and passages from web collection. TREC DL tracks share the same retrieval corpus with MS-MARCO and provide another two annotated test sets. We train our models on the MS-MARCO train set, and then evaluate them on the MS-MARCO development set[4], TREC DL 19 and 20 test sets. Table 1 shows the statistics of these datasets. We evaluate the zero-shot retrieval performance on BEIR benchmark (Thakur et al., 2021), which contains 18 datasets across different domains.

**Evaluation Metrics.** Following previous work, we use MRR@10, Recall@50, and Recall@1000 to evaluate the retrieval performance on MS-MARCO, where MRR@10 is the most important metric. We resort to nDCG@10 for TREC DL

---

[3]We can pre-generate queries and pre-compute document representations. Therefore, the retrieval latency mentioned in this paper does not encompass these pre-processing steps.

[4]The MS-MARCO test set is not publicly available.

tracks and BEIR benchmark[5].

**Implementation Details.** We set the maximum document length to be 144 tokens, and the maximum query length to be 32 tokens. All models are optimized using the AdamW optimizer (Loshchilov and Hutter, 2019) with the learning rate of $5 \times 10^{-6}$, and a linear learning rate schedule with 10% warm-up steps for around three epochs. Every training batch contains 64 queries, and each query is accompanied by one positive and 31 hard negative documents. Following Gao and Callan (2022), we train our proposed model for two stages and initialize the retriever with coCondenser at each stage. At the first training stage, the hard negatives are sampled from the official BM25 hard negatives, but at the second training stage, the hard negatives are sampled from the mined hard negatives. For each training stage, we evaluate the last model training checkpoint on the retrieval datasets.

The query generator is built upon the seq2seq architecture (Sutskever et al., 2014; He and Yiu, 2022), designed to take a passage as its input and has been specifically trained to create synthetic queries that correspond to the provided passage. Nogueira and Lin (2019) initialized the query generator with the T5-base model. They trained it on the MS-MARCO dataset and generated 80 queries for each passage within MS-MARCO. Following Li et al. (2022), we use the queries[6] released by Nogueira and Lin (2019) to expand the passages in the MS-MARCO dataset. In parallel, for BEIR, we use the query generator[7] fine-tuned on MS-MARCO to generate 5 queries for each document within the BEIR benchmark. During the query generation process, we employ the top-$k$ sampling approach with $k$ set to 10, and we limit the maximum length of the synthetic queries to 64 tokens.

During training, we implement our proposed curriculum sampling with all 80 queries and divide them into $K$ groups, but at inference, we resort to the first 10 (i.e., $S = 10$) and first 5 (i.e., $S = 5$) queries to compute the typical document representation for MS-MARCO and BEIR benchmark, respectively. At the first training stage, we set $K$ to 3, and at the second training stage, we set $K$ to 4.

---

[5]The official evaluation toolkit is available at `https://github.com/beir-cellar/beir`.

[6]`https://git.uwaterloo.ca/jimmylin/doc2query-data/raw/master/T5-passage/predicted_queries_topk_sampling.zip`

[7]`https://huggingface.co/castorini/doc2query-t5-base-msmarco`

| Models | MS-MARCO | | | TREC DL 19 | TREC DL 20 |
| | MRR@10 | R@50 | R@1000 | nDCG@10 | nDCG@10 |
|---|---|---|---|---|---|
| **Sparse retrieval** | | | | | |
| BM25 (Yang et al., 2017) | 18.5 | 58.5 | 85.7 | 51.2 | 47.7 |
| DeepCT (Dai and Callan, 2019) | 24.3 | 69.0 | 91.0 | 57.2 | - |
| DocT5Query (Nogueira and Lin, 2019) | 27.7 | 75.6 | 94.7 | 64.2 | - |
| **Dense retrieval** | | | | | |
| DPR (Karpukhin et al., 2020) | 31.4 | - | 95.3 | 59.0 | 62.1 |
| ANCE (Xiong et al., 2021) | 33.0 | - | 95.9 | 64.5 | 64.6 |
| SEED (Lu et al., 2021) | 33.9 | - | 96.1 | - | - |
| STAR (Zhan et al., 2021) | 34.7 | - | - | 68.3 | - |
| TAS-B (Hofstätter et al., 2021) | 34.0 | - | 97.5 | **71.2** | 69.3 |
| RocketQA (Qu et al., 2021) | 37.0 | 85.5 | 97.9 | - | - |
| COIL (Gao et al., 2021) | 35.5 | - | 96.3 | 70.4 | - |
| ColBERT (Khattab and Zaharia, 2020) | 36.0 | 82.9 | 96.8 | - | - |
| DCE (Li et al., 2022) | 36.0 | - | 96.4 | 68.3 | 68.9 |
| RetroMAE (Xiao et al., 2022) | 35.0 | - | 97.6 | - | - |
| Condenser (Gao and Callan, 2021) | 36.6 | - | 97.4 | 69.8 | - |
| coCondenser (Gao and Callan, 2022)* | 37.9 | 86.3 | 98.4 | 70.7 | 69.8 |
| **CAPSTONE** | **38.6** | **86.6** | **98.6** | 71.1 | **70.3** |

Table 2: Passage retrieval results on MS-Marco Dev, and TREC datasets. Results with * are from our reproduction.

During inference, we pre-compute representations for all documents in the retrieval corpus, and build IndexFlatIP indexes for document representations with the FAISS library (Johnson et al., 2019) to accelerate retrieval.

We implement all models with the HuggingFace Transformers library (Wolf et al., 2019) and conduct all experiments on 8 NVIDIA Tesla V100 GPUs with 32 GB memory. We also use the gradient checkpointing technology (Chen et al., 2016) to reduce the memory of deep neural models, and mixed precision training[8] to speed up training.

## 4.2 Experimental Results

**In-domain Performance.** Table 2 shows the performance of our proposed model and baselines on MS-MARCO, TREC-2019 and TREC-2020. Since we initialize our model, CAPSTONE, with coCondenser, coCondenser is the main comparison model. From Table 2, we can see our proposed model improves coCondenser by a large margin, increasing by 0.7 points in MRR@10 on MS-MARCO, 0.4 points on TREC DL 19 and 0.5 points on TREC DL 20 datasets. These improvements verify the effectiveness of our proposed curriculum learning and typical representation strategy.

**Zero-shot Performance.** To test the out-of-domain generalization capabilities, we first use the T5-base model (Nogueira and Lin, 2019) trained on MS-MARCO to generate 5 queries for each document of BEIR benchmark, and then evaluate our

model, CAPSTONE, fine-tuned with MS-MARCO on the BEIR benchmark. During evaluation, we compute the typical representation for each document with $S = 5$. As shown in Table 3, CAPSTONE performs much better than all baselines in the average performance. Compared with its counterpart without document expansion (i.e., coCondenser), our model achieves improvements on 11 out of 18 datasets and increases nDCG@10 by 0.6 points in the average performance. These improvements demonstrate that our approach generalizes well on diverse domains under zero-shot settings.

## 4.3 Ablation Study and Analysis

To analyze our method more directly, experiments in this section are based on the vanilla DPR initialized with the ERNIE-2.0-base model (Sun et al., 2020) without using dense retrieval pre-training strategies, if there is no particular statement.

**Corpus Expansion vs. Document Expansion.** In this paper, we append only one query to a document both at training and inference. As stated in 3.3, corpus expansion will enlarge the original corpus to $S$ times during inference. In contrast, document expansion for sparse retrieval (Nogueira and Lin, 2019) appends multiple generated queries to a document at inference, thus without bringing extra retrieval latency. Since document expansion[9] has shown effectiveness in sparse retrieval, it is very intuitive to consider whether it is effective for dense retrieval. To answer this question, we

---

[8]https://github.com/NVIDIA/apex

[9]In this section, document expansion means expanding a document with multiple queries.

| Dataset | BERT | LaPraDoR | SimCSE | DiffCSE | SEED | Condenser | coCondenser* | CAPSTONE |
|---|---|---|---|---|---|---|---|---|
| TREC-COVID | 64.9 | 49.5 | 52.4 | 49.2 | 61.2 | 75.4 | 74.0 | **77.9** |
| BioASQ | 26.2 | 23.9 | 26.4 | 25.8 | 29.7 | 31.7 | 34.1 | **34.3** |
| NFCorpus | 25.7 | 28.3 | 25.0 | 25.9 | 25.6 | 27.8 | 32.4 | **33.0** |
| NQ | 43.8 | 41.5 | 41.2 | 41.2 | 42.5 | 45.9 | 50.5 | **50.5** |
| HotpotQA | 47.8 | 48.8 | 50.2 | 49.9 | 52.8 | 53.7 | 56.4 | **56.7** |
| FiQA-2018 | 23.7 | 26.6 | 24.0 | 22.9 | 24.4 | 26.1 | 30.0 | **30.4** |
| Signal-1M (RT) | 21.6 | 24.5 | **26.4** | 26.0 | 24.6 | 25.8 | 24.7 | 23.1 |
| TREC-NEWS | 36.2 | 20.6 | 36.8 | 36.3 | 33.5 | 35.3 | 39.1 | **40.3** |
| Robust04 | 36.4 | 31.0 | 35.3 | 34.3 | 34.8 | 35.2 | 40.3 | **40.7** |
| ArguAna | 35.7 | **50.3** | 43.6 | 46.8 | 34.7 | 37.5 | 40.9 | 39.2 |
| Touche-2020 | 27.0 | 17.8 | 17.8 | 16.8 | 18.0 | 22.3 | 27.0 | **31.0** |
| CQADupStack | 28.4 | **32.6** | 29.5 | 30.5 | 28.5 | 31.6 | 30.0 | 30.0 |
| Quora | 78.2 | 84.3 | 84.8 | 85.0 | 84.9 | **85.5** | 84.3 | 83.8 |
| DBPedia | 29.8 | 32.8 | 30.4 | 30.3 | 32.4 | 33.1 | 37.2 | **38.0** |
| SCIDOCS | 11.5 | **14.5** | 12.5 | 12.5 | 11.7 | 13.6 | 14.3 | 14.3 |
| FEVER | 68.4 | 51.8 | 65.1 | 64.1 | 65.3 | 68.2 | 72.4 | **72.7** |
| Climate-FEVER | 20.5 | 17.2 | **22.2** | 20.0 | 17.6 | 19.9 | 19.4 | 19.3 |
| SciFact | 50.4 | 48.3 | 54.5 | 52.3 | 55.6 | 57.0 | 58.3 | **60.5** |
| Avg. Performance | 37.6 | 35.8 | 37.7 | 37.2 | 37.7 | 40.3 | 42.7 | **43.3** |

Table 3: Zero-shot dense retrieval performances on BEIR benchmark (measured with nDCG@10). Results with * are from our reproduction.

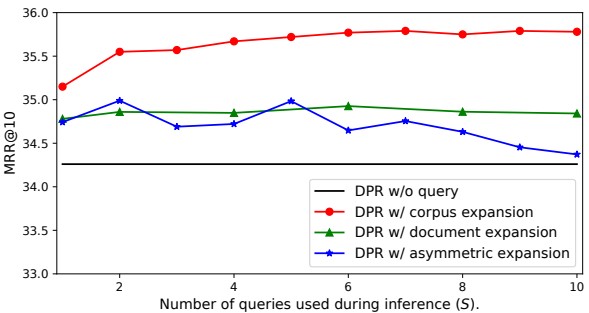

Figure 5: Results of DPR with corpus, document, and asymmetric expansion on the dev set of MS-Marco.

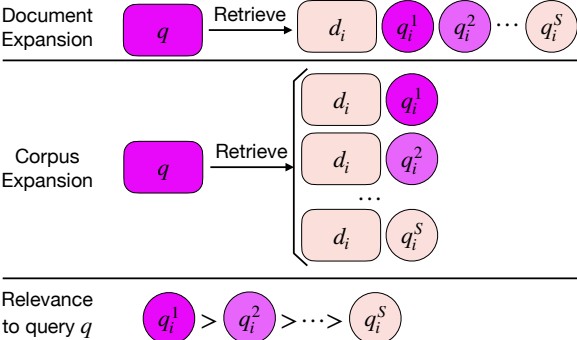

Figure 6: Illustration of document expansion and corpus expansion at inference. For simplicity, only the positive document $d_i$ of the query $q$ is shown.

conduct experiments on DPR with three different settings: (1) corpus expansion; (2) document expansion; (3) asymmetric expansion, which appends one query to a document at training, but appends $S$ queries to a document at inference. As shown in Figure 5, unlike corpus expansion, using more queries does not bring clear improvements for document and asymmetric expansion, but sometimes damages their performances. To explain this, we illustrate document expansion and corpus expansion in Figure 6. Document expansion models the target document and all queries together, where the irrelevant queries acting like noise will corrupt the document representation. By comparison, corpus expansion nicely bypasses this problem by modeling different queries independently.

**Effect of Query Selection Strategies.** To demonstrate the advantage of our proposed curriculum sampling, we expand the positive and negative doc-

uments with different query selection strategies: (1) gold, meaning using the gold query (i.e., strategy used by the vanilla DCE (Li et al., 2022)); (2) random, meaning randomly sampling a query from the generated queries; (3) top-$k$ and (4) bottom-$k$ meaning sampling a query from the top-$k$ and bottom-$k$ sorted queries, respectively.

From Figure 7, we observe that: (1) DPR trained with the gold query performs worst among all strategies, even worse than the vanilla DPR (DPR w/o query) when $S < 8$. As we have stated above, expanding the document with the gold query during training will make the retriever pay more attention to the query while ignoring the document. At inference, when the provided queries are limited (i.e., $S$ is small), it is unlikely to find a query similar to the user-input query. However, as $S$ increases, the

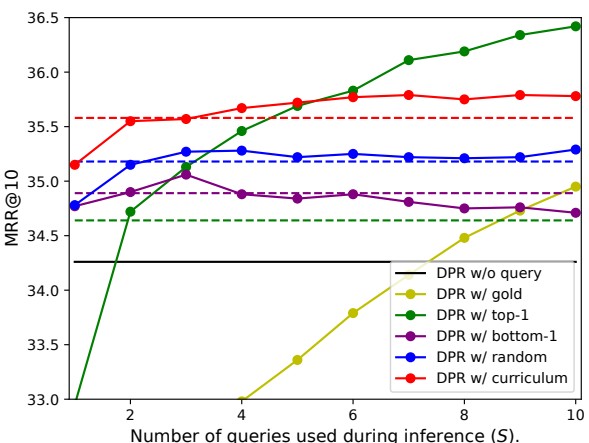

Figure 7: Comparison of different query selection strategies. During training, we first expand the document with a query selected from the generated queries using a specific strategy, and then train DPR with the expanded document. During inference, we evaluate the well-trained DPR retrievers with corpus expansion (solid lines) and the typical representation (dotted lines) on the dev set of MS-Marco, where the solid and dotted lines with the same color are results for the same DPR.

probability of finding a query similar to the user-input query also increases. (2) DPR trained with the top-1 query performs much better than DPR trained with the gold query, since it slightly shifts its attention from the query to the document. However, it still cannot exceed DPR w/o query when $S = 1$, indicating that it fails to fully utilize the document like DPR. (3) If the document is expanded with a weakly related (bottom-1) or random query at training, DPR with both settings will outperform DPR w/o query when $S = 1$. These two strategies enable DPR to mine query information without sacrificing document information. However, their performances are not strongly positively correlated with $S$ like the top-1 strategies, indicating it does not fully use the query information. (4) DPR with the curriculum strategy outperforms DPR w/o query when $S = 1$, and its performance linearly increases with $S$, verifying this strategy enables DPR to utilize both the document and query to model the query-informed document representation.

**Effect of the Typical Representation.** Although corpus expansion makes the retrieval performance of our proposed curriculum sampling improve with the increase of $S$ (see the solid red line in Figure 7), the retrieval latency also increases linearly. To alleviate this, we propose to use the average of different document views as the typical representation. Among all strategies (see the dotted lines in the

| Variants | MRR@10 | R@1000 |
|---|---|---|
| DPR | 34.26 | 97.02 |
| CAPSTONE+DPR w/ $S = 1$ | 35.15 | 97.19 |
| CAPSTONE+DPR w/ average | **35.66** | **97.28** |
| CAPSTONE+DPR w/ max | 35.45 | 97.25 |
| CAPSTONE+DPR w/ median | 35.64 | 97.20 |

Table 4: Results of CAPSTONE initialized with DPR using different pooling methods to compute the typical representation on the dev set of MS-Marco.

| Models | BM25 Negatives | | Mined Negatives | |
|---|---|---|---|---|
| | MRR@10 | R@1000 | MRR@10 | R@1000 |
| DPR | 34.26 | 97.02 | 36.44 | 97.65 |
| CAPSTONE+DPR | 35.66 | 97.28 | 37.28 | 97.82 |
| coCondenser | 35.91 | 98.21 | 37.94 | 98.41 |
| CAPSTONE+coCondenser | **36.75** | 98.21 | **38.65** | 98.60 |

Table 5: Performance of CAPSTONE initialized with DPR and coCondenser at the two training stages on the dev set of MS-Marco.

Figure 7), our proposed curriculum sampling performs best, which again verifies the effectiveness of curriculum sampling. In addition, the retrieval performance of typical representations outperforms its counterpart using corpus expansion with $S = 1$ for all query selection strategies. More importantly, for the curriculum learning strategy, the typical representation enhances the vanilla DPR by 1.2 on MRR10, and is even on par with its counterpart using corpus expansion with $S = 3$, but there still exists a clear performance gap between the typical representation and corpus expansion with $S = 10$. Therefore, the typical representation balances the retrieval efficiency and performance.

**Comparison of Methods for Computing the Typical Representation.** We consider three different pooling methods to compute the typical document representation: taking the average/max/median pooling of different document views. From Table 4, we observe that (1) DPR with $S = 1$ improves DPR by 0.9 on MRR@10; (2) DPR with the typical representation brings an extra 0.5 improvements on MRR@10; (3) There is no significant difference in the results of the three pooling methods. Given the simplicity, we use the average pooling to compute the typical representation.

**Multi-stage Retrieval Performance.** To further verify the effectiveness of our proposed approach, we apply our proposed approach to DPR and coCondenser. From Table 5, we observe that our approach brings clear improvements to DPR with 1.4 and 0.8 increases on MRR@10 at the two training stages, respectively. In addition, we witness similar

improvements on coCondenser, proving that our approach does not depend on retrieval models and is applicable to different retrieval models.

## 5 Related Work

### 5.1 Dense Retrieval

In recent years, with the development of large-scale pre-trained language models, such as BERT (Devlin et al., 2019) and RoBERTa (Liu et al., 2019), we witness the research interest in information retrieval shifting from the traditional sparse retrieval to neural dense retrieval. Karpukhin et al. (2020); Xiong et al. (2021) proposed dense passage retriever (DPR), which uses two neural encoders initialized with BERT to model the query and document, independently. The subsequent works follow this dual-encoder framework. One line of work improves dense retrieval by mining hard negatives, where they select the top-ranked documents retrieved by the recent retriever as hard negatives and then re-train the retriever with the newly mined hard negatives (Xiong et al., 2021; Qu et al., 2021). However, the mined hard negatives are highly likely to contain false negatives, harming the performance. To mitigate this, following studies denoise the mined hard negatives with re-rankers (Qu et al., 2021; Ren et al., 2021; Zhang et al., 2022). Another kind of work focuses on pre-training to make the pre-trained models more suitable for dense retrieval, such as Condenser (Gao and Callan, 2021), coCondenser (Gao and Callan, 2022) and SIMLM (Wang et al., 2022).

Dense retrieval models typically depend on extensive supervised data, comprising pairs of queries and positive documents. To address the challenge of limited training data, Ma et al. (2021); Sun et al. (2021) proposed to train a query generator on high-resource information retrieval data, and then used the query generator to generate synthetic queries for low-source target domains. Additionally, Dai et al. (2023) harnessed large language models in zero/few settings to produce synthetic queries for documents in target domains, eliminating the need for training a general query generator. In contrast to these approaches, our work leverages synthetic queries for document expansion.

### 5.2 Query Expansion

Query expansion enriches the query with various heuristically discovered relevant contexts. For example, GAR (Mao et al., 2021) expands the query by adding relevant contexts, including the answer, the sentence containing the answer, and the title of a passage where the answer belongs. However, query expansion will increase the retrieval latency during inference, since the generation model is required to generate these relevant contexts for query expansion. Therefore, we mainly focus on document expansion in this work.

### 5.3 Document Expansion

Document expansion augments the document with generated queries, which the document might answer. Compared with query expansion, document expansion can be conducted prior to indexing without incurring extra retrieval latency. Document expansion has shown its effectiveness on sparse retrieval models (Nogueira et al., 2019; Nogueira and Lin, 2019), yet how to apply it to dense retrieval models is still under-explored. Li et al. (2022) first applied document expansion to dense retrieval models. However, their models suffer from the discrepancy between training and inference, thus even degrading the vanilla dense retrieval models under the single-view document representation settings. To mitigate this discrepancy, we propose a curriculum learning approach, which successfully applies document expansion to dense retrieval models.

## 6 Conclusion

This work proposes the curriculum sampling for dense retrieval with document expansion, which enables dense retrieval models to learn much better query-related document representations. In addition, we propose to compute the typical representation of different document views, which balances inference efficiency and effectiveness. Our experimental results on the in- and out-of-domain datasets verify the effectiveness of the curriculum sampling and typical representation.

## 7 Limitations

There are two possible limitations of this work. The first limitation is that we need to generate synthetic queries for each document in the retrieval corpus, which is very time-consuming. Luckily, this process does not bring extra delay to retrieval. In addition, limited by sufficient computational resources, we only verify the effectiveness of our method on vanilla DPR and coCondenser. In the future, we plan to apply our approach to other dense retrieval

models and verify the effectiveness of our method on these models.

## Acknowledgements

This project is supported by HKU-SCF FinTech Academy, Shenzhen-Hong Kong-Macao Science and Technology Plan Project (Category C Project: SGDX20210823103537030), and Theme-based Research Scheme of RGC, Hong Kong (T35-710/20-R). We would like to thank the anonymous reviewers for their constructive and informative feedback on this work.

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
