# OpenReview forum: "CAPSTONE: Curriculum Sampling for Dense Retrieval with Document Expansion"
_EMNLP/2023/Conference — EMNLP 2023 Main_

### Official Review · Reviewer_p4HK · 2023-07-31

**Typos Grammar Style And Presentation Improvements:** N/A
**Soundness:** 4

**Excitement:**

4: Strong: This paper deepens the understanding of some phenomenon or lowers the barriers to an existing research direction.

**Missing References:**

Query generation:
[1] Zero-shot neural passage retrieval via domain-targeted synthetic question generation.
[2] Few-Shot Text Ranking with Meta Adapted Synthetic Weak Supervision.
[3] Promptagator: Few-shot Dense Retrieval From 8 Examples.


**Paper Topic And Main Contributions:**

The paper presents a study on the shortcomings of the dual-encoder architecture in dense retrieval systems. While the dual-encoder has become the standard approach, it falls short of capturing the interactions between queries and documents effectively. This paper proposes CAPSTONE, which expands the document with a real query for better representing the documents. However, the training and inference processes may contain exploration bias, which causes by the quality of generated queries. To overcome the challenge, the authors propose a novel curriculum sampling strategy. During training, pseudo queries are introduced to progressively enhance the relevance between the generated query and the real query. This strategy allows the retrieval model to learn and extend its attention from focusing solely on the document to jointly considering both the document and query.

**Questions For The Authors:**

How do you generate document-related queries? And how do you guarantee the diversity of generated queries?

**Reasons To Accept:**

1. This paper proposes a document expansion method, which targets generating document-related queries for document expansion. The idea is intuitive.
2. The experimental results on MS MARCO and BEIR show the effectiveness of the proposed method.
3. The proposed method can alleviate the exploration bias between training and inference, which derives from query generation.

**Reasons To Reject:**

1. Some related work, such as query generation and negative sampling should be discussed.
2. The novelty of this paper aims to alleviate the exploration bias between training and inference due to query generation. However, the query generation method is not jointly optimized.

**Reproducibility:**

4: Could mostly reproduce the results, but there may be some variation because of sample variance or minor variations in their interpretation of the protocol or method.

**Reviewer Confidence:**

3: Pretty sure, but there's a chance I missed something. Although I have a good feel for this area in general, I did not carefully check the paper's details, e.g., the math, experimental design, or novelty.

---

> ### Author Rebuttal · Authors · 2023-08-29
>
> Thanks for your thoughtful and valuable comments.
>
> > *Q1*: Related work about query generation.
>
> We will include the omitted related work concerning query generation in the final version.
>
> > *Q2*: About query generation.
>
> We apologize for not describing this part clearly.
> The query generation model is based on the sequence-to-sequence pipeline, which is initiliazed with the T5-base model. Following Nogueira and Lin (https://cs.uwaterloo.ca/~jimmylin/publications/Nogueira_Lin_2019_docTTTTTquery-v2.pdf), we feed the passage into the generation model and train the model to generate the query. For fair comparisons to DCE (https://arxiv.org/abs/2208.04232), we use the queries released by Nogueira and Lin (https://git.uwaterloo.ca/jimmylin/doc2query-data/raw/master/T5-passage/predicted_queries_topk_sampling.zip) to expand the passages of MS-MARCO.
> In parallel, for BEIR, we use the T5-base model fine-tuned on MS-MARCO to generate 5 queries for each document within the BEIR benchmark. During inference, we use the top-k (k=10) sampling and set the max output length to 64 to generate queries. We will provide a clearer description about query generation in the final version.
>
> Using the query generation model to generate queries can be a time-consuming process. For instance, generating queries for the BEIR dataset took approximately 15 days using 8 V100 GPUs. In light of restricted computational resources, we undertook separate optimization for both the generation model and the retrieval model. In the same vein, we didn't explore impacts of different generation strategies and ways to enhance query diversity. We leave them as future work.

---

### Official Review · Reviewer_4QHG · 2023-08-04

**Soundness:** 4

**Excitement:**

3: Ambivalent: It has merits (e.g., it reports state-of-the-art results, the idea is nice), but there are key weaknesses (e.g., it describes incremental work), and it can significantly benefit from another round of revision. However, I won't object to accepting it if my co-reviewers champion it.

**Paper Topic And Main Contributions:**

The paper deals with the training procedure of dual-cross-encoders for dense retrieval. It builds on the architecture proposed in Li et al (2022) where the dual encoder is composed of a query encoder and a document+query (aka expanded document) encoder. The idea is to enable the document encoder to learn complex relations between query and document (like a cross encoder) while keeping the dual encoder’s reasonable inference time. The novelty over Li et al (2022) includes the following. First, the authors suggest generating queries for the expanded documents during training (and not only during inference), instead of just using the train-set positive queries during training and generating a query only in inference time. Second, In early stages of training authors suggest using queries that are different from the actual positive query and as training proceeds they use queries that are more and more similar to the positive train-set query and 3. To limit the degradation in inference time (caused by the fact that multiple queries are generated for each document in the corpus), they average the different embeddings generated for each document and use only a single embedding per document.
Experiments on several datasets (MS MARCO, TREC 19+20, BEIR) show the method proposed improves upon the baseline.

**Questions For The Authors:**

1. Are the results statistically significant?
2. Do you have any idea why your reproduced coCondenser results are different then the results reported in their paper?

**Reasons To Accept:**

- Paper is well written, figures are informative
- Novel method for training dual-cross-encoders
- Improvement upon baseline training procedure

**Reasons To Reject:**

- I have some concerns regarding the evaluation process: first, are the results statistically significant? specifically, the interesting baseline is of course coCondenser- is CAPSTONE statistically significant better than it? Further, coCondenser’s paper reported MRR@10 of 38.2 on MS MARCO and in this paper the reported score is 37.9, what is the reason for this difference? Finally, it is unclear to me why the main results are based on coCondenser and the ablation study is performed on a vanilla DPR model.

- Authors do not discuss the queries generation process, which in my opinion is a crucial part of the proposed training procedure. I think it can be interesting to study and get some insights into how generating queries differently effects the training process. For most use cases, pre-generated queries will not be available and in order to use the proposed method these queries will need to be generated.

**Reproducibility:**

4: Could mostly reproduce the results, but there may be some variation because of sample variance or minor variations in their interpretation of the protocol or method.

**Reviewer Confidence:**

4: Quite sure. I tried to check the important points carefully. It's unlikely, though conceivable, that I missed something that should affect my ratings.

**Typos Grammar Style And Presentation Improvements:**

line 358: “As shown in Table 3, CAPSTONE performs much than all baselines in the average performance” - much what? :)

line 407: you say you show results on “top-k” and “bottom-k” but the table only shows results for and top-1 and bottom-1

---

> ### Author Rebuttal · Authors · 2023-08-29
>
> Thank you for your thoughtful and invaluable feedback.
>
> > *Q1*: About the evaluation results.
>
> We conducted t-test comparisons between our model and baselines with p-values<0.05, which indicates our results are statistically significant.
>
> We conducted experiments adhering closely to the configuration in coCondenser’s paper, including the utilization of AdamW optimizer with a learning rate of 5e-6, a linear learning rate schedule, and a batch size of 64 for 3 epochs. However, our replication yielded a result of 37.9, whereas the original paper reported 38.2. This discrepancy might stem from variances in experiment random seeds. We set our random seed to 42, whereas they did not mention the selected random seed in their paper.
>
> In the ablation study, we did not use the pre-trained retrieval model, coCondenser, with the aim of directly assessing the effectiveness of our approach. As depicted in Table 5, our method demonstrates its effectiveness even when applied to coCondenser.
>
>
> > *Q2*:  About query generation.
>
> We apologize for not describing this part clearly.
> The query generation model is based on the sequence-to-sequence pipeline, which is initiliazed with the T5-base model. Following Nogueira and Lin (https://cs.uwaterloo.ca/~jimmylin/publications/Nogueira_Lin_2019_docTTTTTquery-v2.pdf), we feed the passage into the generation model and train the model to generate the query. For fair comparisons to DCE (https://arxiv.org/abs/2208.04232), we use the queries released by Nogueira and Lin (https://git.uwaterloo.ca/jimmylin/doc2query-data/raw/master/T5-passage/predicted_queries_topk_sampling.zip) to expand the passages of MS-MARCO.
> In parallel, for BEIR, we use the T5-base model fine-tuned on MS-MARCO to generate 5 queries for each document within the BEIR benchmark. During inference, we use the top-k (k=10) sampling and set the max output length to 64 to generate queries. We will provide a clearer description about query generation in the final version.
>
> Using the query generation model to generate queries can be a time-consuming process. For instance, generating queries for the BEIR dataset took approximately 15 days using 8 V100 GPUs. In light of restricted computational resources, we didn't explore impacts of different generation strategies. We leave them as future work.
>
> > *Q3*:  About typos
>
> We will fix typos and grammar issues in the final version.

---

### Official Review · Reviewer_1aty · 2023-08-06

**Soundness:** 4

**Excitement:**

3: Ambivalent: It has merits (e.g., it reports state-of-the-art results, the idea is nice), but there are key weaknesses (e.g., it describes incremental work), and it can significantly benefit from another round of revision. However, I won't object to accepting it if my co-reviewers champion it.

**Paper Topic And Main Contributions:**

The paper proposes a new approach for representing documents and queries to enable dense retrieval. The approach aims to generate synthetic queries for retrieving documents so that the model learns better representations of documents. This is done by discouraging the modeling of trivial associations between queries and documents within a batch, which the authors argue that current dual cross-encoders tend to do. Overall, the approach is simple and intuitive, and results on two existing datasets demonstrate the approach's effectiveness over several competitive baselines.

**Questions For The Authors:**

I would like to thank the authors for this work. I appreciate how the authors carefully highlight the limitation of the dual cross-encoders used for dense retrieval and then provide a simple but well-thought approach to counter them. The results are encouraging! I have the following comments and questions about the work.

A. The process of generating synthetic queries is unclear to me. From the appendix, I understand that the authors used the queries provided by Nogueira and Lin (2019) for MS-MARCO. It'd be beneficial to include an overview of the approach they adopt. It was unclear how the authors fine-tuned a T5-based model to obtain the queries for documents in the BEIR benchmark. Furthermore, could the authors provide qualitative examples for the synthetic queries from each of the groups in curriculum learning – what do the queries that have low relevance with gold query (q) look like, and how do they compare to those with high relevance with q?

B. The argument about no delay in retrieval time is unclear (mentioned in lines 554-557 and other places earlier). The authors propose an approach that generates multiple queries for each document, which are aggregated before final retrieval. Does generating multiple synthetic queries not add latency to the retrieval process? Why do the authors claim no additional latency?

C. I would request authors to note which ROUGE implementation they used as a recent ACL'23 work demonstrates the discrepancy and bugs in ROUGE-based evaluation for an overwhelming number of published papers. Please see Grusky et al. (ACL 2023: https://aclanthology.org/2023.acl-long.107.pdf) for more details and pointers to ensure the ROUGE implementation they use is correct.

D. What do 144 and 32 mean in lines 746-747? Are the maximum lengths for documents and queries set based on tokens, words, or character counts?

**Reasons To Accept:**

- The approach is simple and intuitive, and the authors clearly articulate the motivation behind the approach. Overall, I found the contributions of this work to be well-motivated
- The results demonstrate the effectiveness of the proposed approach over several baselines on multiple datasets

**Reasons To Reject:**

- The approach used to construct the synthetic queries is unclear to me. The authors defer the reader to information in the appendix, which doesn't provide sufficient information. The construction of synthetic queries is one of the key steps of the proposed approach
- The authors argue that the existing dual cross-encoding approaches tend to disregard in-batch negatives while learning. This would have been a more compelling argument if the authors could provide empirical evidence. For instance, do the visualization of weights provide evidence for heterogeneity between training- and inference times?

**Reproducibility:**

4: Could mostly reproduce the results, but there may be some variation because of sample variance or minor variations in their interpretation of the protocol or method.

**Reviewer Confidence:**

3: Pretty sure, but there's a chance I missed something. Although I have a good feel for this area in general, I did not carefully check the paper's details, e.g., the math, experimental design, or novelty.

---

> ### Author Rebuttal · Authors · 2023-08-29
>
> Thanks for your thoughtful and valuable comments.
>
> > *Q1*: About query generation.
>
> We apologize for not describing this part clearly.
> The query generation model is based on the sequence-to-sequence pipeline, which is initiliazed with the T5-base model. Following Nogueira and Lin (https://cs.uwaterloo.ca/~jimmylin/publications/Nogueira_Lin_2019_docTTTTTquery-v2.pdf), we feed the passage into the generation model and train the model to generate the query. For fair comparisons to DCE (https://arxiv.org/abs/2208.04232), we use the queries released by Nogueira and Lin (https://git.uwaterloo.ca/jimmylin/doc2query-data/raw/master/T5-passage/predicted_queries_topk_sampling.zip) to expand the passages of MS-MARCO.
> In parallel, for BEIR, we use the T5-base model fine-tuned on MS-MARCO to generate 5 queries for each document within the BEIR benchmark. During inference, we use the top-k (k=10) sampling and set the max output length to 64 to generate queries. We will provide a clearer description about query generation in the final version.
>
> For ease of understanding, we provide an example of the synthetic queries from each of the groups in curriculum learning.
> During training, suppose the gold query is "who proposed the geocentric theory". The synthetic queries in different groups are shown as follows (for simplicity, we only show one query for each group): \
> Group 1: who proposed the geocentric model \
> Group 2: who discovered the geocentric system of plato\
> Group 3: what was plato's geocentric theory of the planets\
> Group 4: what model did plato use to determine retrograde motion
>
> The queries from group 1 are most relevant to the gold query, while the queries from group 4 are the least relevant to the gold query.
>
>
> > *Q2*:  About no delay in retrieval.
>
> We apologize for not stating this clearly. We will provide a clearer description of this issue in the future version. What we are referring to is that before using the capstone model (our model), we need to generate queries for each document using the generation model, and this step is quite time-consuming. However, since we can pre-generate queries, the capstone model does not introduce any additional retrieval latency compared to the vanilla dense retrieval models at the retrieval stage.
>
>
> > *Q3*:  About ROUGE-L
>
> Thank you for your thoughtful reminder. We utilize the ROUGE package from the HuggingFace Hub to compute ROUGE-L (https://huggingface.co/spaces/evaluate-metric/rouge). This package is a wrapper around Google Research's reimplementation of ROUGE (https://github.com/google-research/google-research/tree/master/rouge). We intend to incorporate this information into the implementation details for the future version. It is worth noting that we only utilize ROUGE-L to rank all candidate queries, rather than employing it as an evaluation metric. Therefore, the reproducibility issues highlighted in the ACL paper due to variations in the calculation of ROUGE are not applicable here.
>
>
> > *Q4*: About the maximum lengths.
>
> In lines 746-747, the maximum lengths for documents and queries are set based on tokens, where the inputs are tokenized with the huggingface tokenizers. We will provide a clearer description in the final version.

---

### Meta-Review · Area_Chair_mvu8 · 2023-09-22

**Recommendation:** 5

**Metareview:**

All the reviewers have agreed that this paper has tackles a pain in dual-cross-encoders.

However, some concerns have also been raised such as some important details are lacking and generally during the rebuttal period, the authors have provided these details and addressed these concerns.

I recommend the authors incorporate these details to the paper.

---

### Decision · Program_Chairs · 2023-10-07

**Decision:**

Accept-Main

**Comment:**

All the reviewers have agreed that this paper has tackles a pain in dual-cross-encoders.

However, some concerns have also been raised such as some important details are lacking and generally during the rebuttal period, the authors have provided these details and addressed these concerns.

I recommend the authors incorporate these details to the paper.